# The Growing Altitude Influences the Flavor Precursors, Sensory Characteristics and Cupping Quality of the Pu’er Coffee Bean

**DOI:** 10.3390/foods13233842

**Published:** 2024-11-28

**Authors:** Rongsuo Hu, Fei Xu, Xiao Chen, Qinrui Kuang, Xingyuan Xiao, Wenjiang Dong

**Affiliations:** 1Spice and Beverage Research Institute, Chinese Academy of Tropical Agricultural Sciences (CATAS), Wanning 571533, China; hnhrs@126.com (R.H.);; 2College of Food and Technology, Nanjing Agriculture University, Nanjing 210095, China; xiaochen@njau.edu.cn; 3Key Laboratory of Processing Suitability and Quality Control of the Special Tropical Crops of Hainan Province, Wanning 571533, China; 4College of Modern Agriculture and Bioengineering, Yangtze Normal University, Chongqing 408100, China; kk2698591303@163.com; 5College of Tropical Crops, Yunnan Agriculture University, Pu’er 665000, China

**Keywords:** coffee, altitude, flavor precursors, sensory characteristics, cupping quality

## Abstract

The growing altitude is an important factor affecting the quality of coffee. We explored the flavor precursors, sensory characteristics, and cupping qualities of coffee growing at different altitudes and discussed their associated relationships. The altitude at which coffee is grown has different effects on its chemical composition. Fatty acid contents increased with increasing altitudes, whereas alkaloid and chlorogenic acids contents decreased with increasing elevation. There was no obvious trend in either organic acids or monosaccharides. Eleven of the 112 detected volatile components were significantly affected by the growing altitude. The contents of pyrazines and alcohols tended to decrease, whereas those of aldehydes tended to increase. A significantly altered composition reduces the nutty and roasted flavors of coffee, while increasing the sweet sugar and caramel aromas. The aroma and flavor tended to increase with increasing altitudes during cupping, whereas the other indicators did not change significantly. The results provide a theoretical reference for the sales and promotion of Pu’er coffee.

## 1. Introduction

Coffee, tea, and cocoa are recognized as the three primary beverages globally. Coffee is favored for its distinctive and appealing flavor and taste [1]. Currently, the United States remains the largest coffee consumer, with the European Union, Brazil, Japan, and Canada following. China has emerged as a new coffee consumer, with both the number of consumers and the volume of consumption rising significantly. Over the past decade, coffee has exhibited an average annual growth rate of 15%.

The rapid increase in coffee consumers has directly resulted in a significant rise in coffee sales, which has heightened the desire among certain consumers for high-quality coffee [2]. The demand for superior flavor and taste has also contributed to the growth of the planting and processing sectors. More than 25 million individuals are involved in coffee cultivation and processing globally. The cultivation and processing of coffee has also yielded substantial benefits for communities in this area [3].

China can also produce high-quality coffee after nearly a century of development. Pu’er City is one of the representative producing areas. This area has a typical hot and humid climate, with morning fog and large day–night temperature differences, which are very suitable for coffee planting and growth. However, Pu’er City is hilly, and coffee is cultivated at altitudes ranging from 900 m to 1600 m. The influence of different altitudes on coffee flavor and quality remains unclear [4].

The altitude at which coffee is cultivated has been shown to be one of the factors affecting the quality of coffee [5]. It is generally believed that coffee grown at less than 900 m is called low-altitude coffee, 900–1200 m is called medium-altitude coffee, 1200–1500 m is called high-altitude coffee, and coffee grown at more than 1500 m is called super high-altitude coffee. The altitude of coffee is often mentioned in terms of production, trade, roasting, brewing, and sales. An adequate altitude determines coffee quality [6]. Therefore, some countries even mark the growth altitude on packaging bags in the coffee trade.

Although the cultivation of coffee at high altitudes positively influences the final quality of the beverage [7], few researchers have connected the final quality to the growth altitude. The sensory profile of coffee produced at various times within the same region may differ. Consequently, studies examining the effects of altitude on flavor characteristics may lack representativeness. Despite variations in climatic conditions over different periods, sampling at the same time can reduce errors associated with the climate’s impact on coffee quality [8].

The quality of coffee is closely related to the precursor components. This was described by researchers in previous studies. The precursor components include both chemical components that affect flavor and quality, such as trace elements [9], alkaloids, and amino acids [10], as well as nutritional components such as organic acids, monosaccharides, and fatty acids [11]. The chemical components mainly directly affect the quality of coffee cups. There are also a small number of components that react with other components in the brewing part, indirectly affecting the quality of coffee. The nutritional components of coffee beans mainly produce the Maillard reaction in the roasting process, forming the aroma component and contributing to the quality of coffee.

Coffee from Pu’er City in Yunnan Province was chosen as the subject of this research. Coffee samples were collected from the same area at altitudes ranging from 930 to 1520 m to analyze the flavor precursor and volatile components. Additionally, a professional team was formed to assess the cupping quality and evaluate the impact of growing altitude on sensory quality. This study aimed to investigate the effects of altitude on the composition of raw coffee beans and the sensory and cupping characteristics of roasted coffee, thereby providing valuable insights for potential applications.

## 2. Materials and Methods

### 2.1. Sample Collection and Processing

#### 2.1.1. Green Coffee Preparation

Coffee cherries were collected in Pu’er City, which is located at a range of altitudes (930–1520 m asl). They were drawn from three hills of a village (Dakaihe village, Nanping town, Simao district, Pu’er City, Yunnan Province, Southwest China). The coffee sample is Coffea arabica L and the variety is catimor CIFC7963 (F6). The samples were collected at intervals of about 150 m, which can cover the growing range in the Pu’er producing area and highlight the altitude differences between samples. The specific sampling sites were as follows: sample A (930 m asl, the Tanfang region), sample B (1100 m asl, the Donggualin region), sample C (1250 m asl, the Xiaka region), sample D (1370 m asl, the Xiaka region), and sample E (1520 m asl, the Xiaka region). The coffee samples were collected at the harvest peak, when the coffee cherry was of better quality. Coffee cherries were subsequently processed via wet processing on the same day and then subjected to solar drying [5]. When the moisture content reached about 12%, the coffee beans were sealed, stored, and shelled when used.

#### 2.1.2. Roasted Coffee Preparation

The roasted coffees were prepared using a coffee roaster (Probatino, Probat Werke von Gimborn Maschinenfabrik GmbH, Emmerich am Rhein, Germany). The roaster was preheated for half an hour prior to sample preparation and roasted 3 times with the same coffee beans to stabilize the equipment while cleaning the oven. The coffee samples were added when the oven was at 175 °C and poured out when the oven was at 195 °C. The fire power of the roaster remained constant throughout the roasting process, and the roasting time lasted between 618 and 628 s [12,13].

### 2.2. Chemical Composition Analysis of Green Coffee Beans

#### 2.2.1. Trace Element Determination

The samples were ashed in a KSL-1400X-A1 muffle furnace (Kuncheng Scientific Instrument Co., Ltd., Shanghai, China). The trace elements were subsequently detected with a Z-2310 atomic absorption spectrophotometer (Hitachi Ltd., Chiyoda-ku, Tokyo, Japan). The potassium content was measured with a flame photometer, whereas the manganese, magnesium, calcium, ferrum, cuprum and zinc contents were detected with an atomic absorption spectrophotometer. The detection parameters were as follows: the determination signal was BKG correction, the signal calculation was integrated, the gas pressure was 160 kPa, the auxiliary gas flow rate was 15.0 L/min, the burner height was 7.5 mm, the delay time was 5 s, and the calculation method was the standard curve method. The element detection wavelengths were as follows: calcium, 422.7 nm; magnesium, 285.2 nm; cuprum, 324.8 nm; zinc, 213.9 nm; manganese, 279.5 nm; and ferrum, 248.3 nm [14].

#### 2.2.2. Alkaloid and Chlorogenic Acids Determination

Caffeine, trigonelline, and chlorogenic acids (CGAs) were analyzed using a 1290 high-performance liquid chromatography (HPLC) system (Agilent Technologies Inc., Palo Alto, CA, USA). Sample preparation was carried out according to previously established methods, with minor modifications [15]. Caffeine, trigonelline, and CGAs were detected using the same method. Separation was achieved using an Eclipse XDB-C18 column (4.6 mm × 250 mm, 5 μm particle size) at a flow rate of 0.5 mL/min and a column temperature of 28 °C. The mobile phase consisted of 0.1% formic acid in water, and the injection volume was 2 μL. The detector wavelengths were set at 275 nm for caffeine, 268 nm for trigonelline, and 254 nm and 325 nm for CGAs.

Six different configurations of CGAs were detected in this experiment. The standard products for alkaloids and CGAs were purchased from Shanghai Yuanye (Shanghai Yuanye Biotechnology Co., Ltd., Shanghai, China). The exact content of the samples was determined by external standard method. Based on the reference material, the standard curve was drawn by external standard method to ensure the accurate quantification of the sample.

### 2.3. Nutrient Composition Analysis of Green Coffee Beans

#### 2.3.1. Organic Acid Determination

The samples were prepared using established laboratory techniques [15] and analyzed with an HPLC system featuring a Zorbax SB-Aq analytical column (4.6 mm × 250 mm, 5 μm particle size). The mobile phase was composed of a methanol and 0.01 mol/L NaH_2_PO_4_ buffer solution (pH = 2.00 ± 0.02, adjusted with o-phosphoric acid). An isocratic elution program was executed with the mobile phases at a flow rate of 0.5 mL/min and a column temperature of 28 °C. A detection wavelength of 210 nm was utilized, and an injection volume of 10 μL was applied. The quantification of organic acids was conducted through a standard curve, with the data expressed as g/100 g DW of coffee beans.

#### 2.3.2. Fatty Acid Determination

The samples were first subjected to chloroform-methanol–methyl esterification and then detected by gas chromatography mass spectrometry (GC-MS), which was equipped with a CP-Sil88 FAME chromatographic column (7890A-5975C, Agilent Technologies Inc., Palo Alto, CA, USA). The contents of fatty acids were determined using the external standard method, and the reference standard was 37 fatty acid standard mixtures of methyl esters (Supelco Inc., Bellefonte, PA, USA). The detection parameters were as follows: the initial temperature started at 130 °C, increased to 180 °C at 10 °C/min, held for 6 min, increased to 195 °C at 1 °C/min, held for 20 min, increased to 225 °C at 2 °C/min, and held for 11 min. Helium was used as the carrier gas at a flow rate of 1.0 mL/min. The injector volume was 2 µL. The injector temperature, ion source temperature, and transmission line temperature were 240 °C, 230 °C, and 280 °C, respectively. Sample detection was conducted in full-scan mode at 30–400 *m*/*z* [15].

#### 2.3.3. Monosaccharide Determination

The coffee samples were hydrolyzed with trifluoroacetic acid and then subjected to acetylation, which was estimated according to the method described, with some modifications. The monosaccharide content was detected using a GC-MS analysis system equipped with a DB-5MS column (30 m × 250 µm, 0.25 μm particle size). The detection parameters were as follows: initial temperature, 50 °C; hold for 3 min; increase to 150 °C at 10 °C/min; hold for 3 min; increase to 220 °C at 4 °C/min; hold for 3 min; and finally increase to 280 °C at 10 °C/min. The injector volume was 1 µL. The injector temperature, ion source temperature, and transmission line temperature were 240 °C, 230 °C, and 280 °C, respectively. Sample detection was conducted in full-scan mode at 25–300 *m*/*z*. The quantitative method was an external standard method with 3-octanol as the standard. The default response factor for each compound to 3-octanol was 1. It was used to calculate the content of each compound.

### 2.4. Flavor Analysis of Roasted Coffee

The volatile compounds of the roasted samples were analyzed using GC-MS, which included a headspace solid phase microextraction component (PAL RSI 85, CTC Analytics, Zwingen, Switzerland). A precisely weighed 1.0 g sample and 0.3 g sodium chloride were placed in 20 mL screw vials sealed with polytetrafluorethylene silicone septa. The samples were incubated at 60 °C for 10 min and adsorbed for 30 min using a 50/30 µm DVB/CAR/PDMS fiber provided by Anpel (Anpel Laboratory Technologies, Shanghai, China). Following extraction, the fiber was promptly inserted into the injection port of the GC-MS for desorption at 250 °C for 5 min [16].

Chromatographic separation was carried out on a DB-WAX capillary column (30 m × 0.25 mm × 0.25 μm). The analytical parameters included an initial oven temperature of 40 °C maintained for 2 min, followed by a ramp to 100 °C at 13 °C/min, an increase to 200 °C at 3 °C/min, and a further rise to 230 °C at 8 °C/min. Helium was employed as the carrier gas, with a flow rate of 1 mL/min. The injector and transfer line temperatures were both set at 250 °C. Detection occurred in full-scan mode within the mass range of *m*/*z* 35–300 [17]. The qualitative analysis of unknown compounds was performed via mass spectrometry, with comparisons made to the NIST 17 standard library. The quantitative analysis utilized an external standard method, with 3-octanol serving as the standard, and a default response factor of 1 was assigned to each compound relative to 3-octanol for calculating the content of each compound.

### 2.5. Sensory Analysis of Roasted Coffee

#### 2.5.1. Electronic Sensory Analysis

E-nose detection was conducted using the Gemini olfactory analysis system. This system comprises pattern recognition software, an HS-100 autosampler, and a sensory array unit. The sensory array includes six metal oxide semiconductor sensors (T30/1, T70/2, PA/2, P30/2, LY2/AA, and LY2/gCT) (Alpha M. O. S., Toulouse, France). Air was utilized as the carrier gas at a flow rate of 150 mL/min. Samples were incubated prior to injection and held at 80 °C for 5 min (300 rpm). The sample injection volume was 1500 μL [17].

#### 2.5.2. Cupping Quality Analysis

Coffee samples were distributed to three different teams, comprising a total of 9 Q-Grande coffee connoisseurs for sensory evaluation. Each team was from a distinct unit: the first team represented Guodian Coffee Company, the second team was affiliated with Yunnan Agricultural University, and the third team was from the Jinglan Coffee training school established by scientific research institutes. The samples were roasted within 6 days prior to evaluation [18]. Subsequently, 13.75 g of coffee powder was placed in five cups, and about 250 mL of clean, odor-free hot water (about 93 °C) was added to the coffee powder [5]. The cupping process was conducted following the operational procedures outlined by the SCA [19].

### 2.6. Statistical Analysis

Statistical analysis was performed through one-way analysis of variance (ANOVA), followed by Duncan’s multiple test to identify significant differences (SPSS 22.0, SPSS Inc., Chicago, IL, USA). Origin 2022 (OriginLab Corporation, Northampton, MA, USA) was used for data plotting. Orthogonal partial least squares discriminate analysis (OPLS-DA) was performed using SIMCA software (version 14.1; Umetrics, Umeå, Sweden). The data are presented as the means ± standard deviations, and each set of data was repeated three times.

## 3. Results and Discussion

### 3.1. Chemical Composition

#### 3.1.1. Trace Elements

Trace elements are essential components of coffee and could play an important role in the normal functioning of the body [20]. The contents of seven trace elements in the coffee beans were determined via atomic absorption spectrometry, and the results are shown in Figure 1. The highest content of trace elements was potassium, followed by magnesium and calcium. The contents of manganese, ferrum, cuprum, and zinc were relatively low. Similar findings were also reported by Cuong et al. in their analysis of Vietnam coffee [9].

The contents of potassium, calcium, and magnesium decreased with increasing planting altitudes. Potassium significantly decreased from 1370 m to 1530 m, whereas magnesium significantly increased at 1370 m. Manganese initially decreased, but then increased. Ferrum, copper, and zinc were largely unaffected by the growing altitude, and their contents did not change significantly with increasing growing altitudes. The concentration of trace elements in the coffee increased from 3.5% to 4.0% (*w*/*w*) during the roasting process. These elements were extracted into the coffee beverage, impacting its flavor profile [21]. Elevated levels of potassium, calcium, and magnesium could have adverse effects on taste [21]. Their concentration decreased with increasing elevation, which might be one reason for the high quality of high-altitude coffee.

#### 3.1.2. Alkaloids and Chlorogenic Acids

Caffeine is the most famous alkaloid in coffee. It is responsible for stimulation, significantly increases its bitter taste [22], and has certain health benefits [23]. The caffeine content is normally distributed with increasing altitude. The highest content at 1100 m was 1.82 g/100 g, and the lowest content at 1520 m was 1.19 g/100 g. There was a significant difference in caffeine content from 1370 m to 1520 m, but there was no significant difference at the other growing altitudes. The caffeine content varied significantly between 1370 m and 1520 m, but not among the other depths. Therefore, high-altitude coffee had a reduced caffeine content and reduced overall complexity of the coffee flavor. Trigonelline is another alkaloid in coffee that has a slightly bitter taste. The content of trigonelline fluctuated with increasing altitudes. Although the content varied, the overall difference was not significant.

Chlorogenic acids (CGAs) are thermally unstable phenolic compounds that significantly influence the final cupping quality and health benefits [24]. This study determined the CGA content of six different configurations, as shown in Figure 2. The concentrations of these six CGAs varied, generally decreasing with increasing altitudes. This finding was the opposite to the results reported by Martins et al. [10]. Trigonellines, 4-CQA, and 3,5-diCQA were found to be positively correlated with coffee astringency and aftertaste astringency [25,26]. Additionally, 3,4-diCQA was linked to sweetness and satiety. This reduction in astringent substances could have a positive effect on coffee quality, but it did not show up in coffee cupping. This could be the reason for some interactions between these compounds and other chemical components in coffee [27].

### 3.2. Analysis of Flavor Precursors for Coffee Beans Grown at Various Altitudes

#### 3.2.1. Organic Acids

Organic acids are related to coffee acidity, and the structural composition and content differences have a direct effect on the cupping quality [10]. Five organic acids were detected in the raw coffee beans, as shown in Table 1. Quinic acid was the main organic acid, followed by malic acid, oxalic acid, acetic acid, and citric acid. There was no obvious change trend in organic acid content with increasing growing altitudes. Quinic acid forms chlorogenolactone with CGAs during the roasting process, increasing the bitterness of the beverage [28].

Malic acid and citric acid are ideal for enhancing the perception of sweetness, flavor freshness, and fruitiness [29]. At an altitude of 1100 m, the concentration of malic acid rose dramatically, reaching 3.77 times that at 930 m and 4.21 times that at 1250 m. The amount of malic acid decreased during the roasting process [3]. When its residual concentration was too high, a distinct bitterness was perceived in the cupping test [30]. Citric acid was also significantly affected by altitude, with its content decreasing sharply beyond 1100 m. Furthermore, citric acid was decreased during roasting and might not be present in high-altitude coffee.

Acetic acid exhibited a trend similar to that of citric acid. However, its concentration increased during roasting [3], imparting fruity, wine-like, and fermented aromas to the coffee beverage [31]. Oxalic acid behaved similarly to quinic acid, showing no clear trend with growing altitudes. It is a relatively mild acid that enhances flavor and has a minimal impact on coffee [27].

#### 3.2.2. Fatty Acids

Fatty acids are important compounds related to texture and taste and play an important role in the sensory experience of coffee. They facilitate cream formation, enhanced taste, and carry volatile compounds that affect aroma and flavor [32]. In particular, fatty acids participate in the Strecker degradation reaction during the roasting process to produce new flavor compounds, which also increase the degree of taste [33]. The composition and content of fatty acids at five different altitudes are shown in Table 1. The content of fatty acids changed regularly with increasing growing altitudes, with an overall trend of increasing first and then decreasing.

The content of linoleic acid was the highest, followed by palmitic acid, stearic acid, and oleic acid. The contents of arachidic acid and linolenic acid were the lowest. Similar results existed in other reports [11]. The content growth of the six fatty acids did not significantly differ from 930 m to 1100 m but increased significantly from 1100 m to 1250 m. Palmitic acid, with the smallest increase, also reached 183% of the original content. In addition to palmitic acid, the fatty acid content increases significantly from 1250 m to 1370 m. The contents of linoleic acid, palmitic acid, and stearic acid decreased significantly from 1370 m to 1520 m, while the contents of oleic acid, arachidonic acid, and linolenic acid decreased, but the difference was not significant. In summary, high altitudes increased the coffee fatty acid content, whereas altitudes that are too high had the opposite effect.

#### 3.2.3. Monosaccharide

Carbohydrates, mainly cellulose, hemicellulose, starch, and a small amount of glycogen, constitute an important part of coffee. The presence of these components had a relatively small effect on the sensory quality of the coffee. Notably, they also underwent a small amount of degradation and other reactions in the roasting process, which could have a certain effect on coffee beverages. Monosaccharides are carbohydrate components that can be directly sensed in coffee. The monosaccharide components at different elevations are shown in Table 1. Five monosaccharides, namely, L-(+)-threose, DL-arabinose, D-(+)-talose, D-(+)-galactose, and D-(+)-mannose, were detected. D-(+)-Talose was the most abundant monosaccharide fraction, followed by D-(+)-mannose and DL-arabinose.

The contents of several monosaccharides did not significantly change with increasing growing altitudes. The monosaccharide content was greater at both ultra-low altitudes and ultra-high altitudes than at other altitudes. Higher levels of monosaccharides made the coffee beverage sweeter. The monosaccharides also reacted with amino acids (Maillard reaction) to produce colored compounds that provided the ideal color formation for coffee beverages. Volatile compounds were produced in these reactions, which had a great impact on the aroma of the coffee and improved its quality [34].

### 3.3. Flavor Compounds in Coffee Beans Grown at Various Altitudes

#### 3.3.1. Volatile Compounds as Analyzed by GC-MS

The volatile aroma of coffee is one of the most complex sensory traits. It is formed by many volatile compounds with different aroma qualities, strengths, and concentrations [35]. A total of 112 volatile compounds were detected in the roasted coffee at different growing elevations (Appendix A). Their contents were 33.00 mg/g, 29.47 mg/g, 29.57 mg/g, 26.49 mg/g, and 26.47 mg/g, respectively. The volatile composition of the coffee gradually decreased with increasing growing altitudes.

The variables with a VIP (variable importance in the projection) greater than 1 were considered important factors contributing to group discrimination, and a higher VIP score indicated better discrimination [36]. VIP was used to screen for components whose abundance significantly changed with increasing altitudes, and the results are shown in Table 2. Eleven compounds were significantly affected by the growing altitude (VIP > 1), nine of which tended to decrease with increasing elevation and two of which tended to increase. The contents of these 11 compounds accounted for 58.54%, 54.41%, 53.92%, 48.71%, and 47.46% of the total content of the volatile components, respectively.

Pyrazine was the compound most affected by the growing altitude, with seven of the nine compounds showing a decreasing trend. Pyrazine was the main volatile component in Pu’er coffee, and a total of 24 species were detected, which provided a nutty and roasted aroma for Pu’er coffee, increasing the complexity and depth of the aroma. The other two compounds that significantly decreased in content with increasing growth altitudes were 2-furanmethanol and pyridine. 2-Furanmethanol was the largest alcohol compound in Pu’er coffee and is a milder compound that was once considered the main source of the sweet caramel-flavored aroma [37]. Pyridine is a pungent and diffuse compound. The reduction in pyridine in high-altitude coffee had a positive effect on the coffee flavor.

The two components that increased with increasing growth altitudes were all aldehydes, namely, furfural and 5-methyl-2-furancarboxaldehyde. Furfural has been described as pungent but sweet, bread-like, caramellic, and cinnamon-almond-like. Many reports indicated that these compounds had a relatively high content profile, and both had a positive influence on the flavor of coffee [38]. In addition to the 11 components with significant elevation effects, there were many nonsignificant components. For example, ketones, furans, phenols, and other components contributed sweet, caramel, and burnt aromas, as well as fruit aromas, to Pu’er coffee. These flavors also constituted the main flavors of Pu’ er coffee. The significantly different flavor components decreased the nutty and roasted flavors of the Pu’er coffee while increasing the sweet and caramel aromas [39].

#### 3.3.2. Electronic Sensory Evaluation

Electronic sensing was conducted using an electronic nose, which provided a comprehensive evaluation of the flavor profile of the coffee powder while minimizing artificial errors. The volatile composition was analyzed to identify differences in specific flavor components, allowing for mutual verification. The results from the electronic sensing are illustrated in Figure 3. The fit indices for the independent variable (R^2^x), the dependent variable (R^2^y), and the model prediction index (Q2) were 1.000, 0.626, and 0.187, respectively. Both the R^2^X and R^2^Y values exceeded 0.5, indicating a satisfactory fit of the predicted data and reasonable accuracy in data visualization [40].

The low value of Q2 indicated that although each sample could be distinguished, the difference was not significant. After 200 permutation tests, the intersection point of the Q2 regression line and the vertical axis was −0.769, indicating that there was no over fitting in the model and that the model verification was effective. T70/2 and PA/2 could be identified by the VIP as the key sensors that form the difference. Moreover, the principal composition analysis and cluster analysis revealed differences in flavor between high-altitude coffee and low-altitude coffee.

#### 3.3.3. Sensory Evaluation of Coffee Cupping

Coffee cupping is a widely accepted technology utilized globally, despite its inherent subjectivity [1]. Research had sought to establish connections between the physicochemical properties and the quality of beverages, aiming to enhance sensory analysis in a more standardized and precise manner [1]. Nine certified tasters participated in the evaluation to ensure more reliable results. All samples received a score of 10 for uniformity, clean cup, and sweetness. The scores for the other attributes and the total score are presented in Figure 4.

The altitude did not affect any of the indicators. The aroma index varied regularly with increasing altitudes, decreased slightly from 930 m to 1100 m, and then reached the highest average score of 7.39 at 1530 m. Flavor showed an increasing trend with increasing elevation, with a maximum value of 7.75 at 1530 m. Body, balance, acidity, and aftertaste all had good ratings but differed little between the planting elevations and were hardly affected by elevation. Overall, a relatively special evaluation, which was based on other indicators, can have certain personal preferences. Overall, there was also a slightly increasing trend. The total score was influenced by the aroma, flavor, and total score, which also tended to increase. All the samples obtained a final score greater than 81 points, classifying them as excellent specialty coffees [19].

There was an obvious controversy in the score of all the index evaluations at 1100 m. This might be because the average altitude of the sample collection site was about 1100 m, and long-term planting led to a decline in soil fertility and a decrease in coffee quality [41]. It might also be that the evaluator is familiar with the coffee growing at this altitude and had a slight dietary preference [42].

### 3.4. Correlation Analysis of Effects of Flavor Precursors and Volatile Components on Cupping Quality

The quality of coffee cupping was closely related to the flavor precursor components and volatile components. The VIP was selected to identify the chemical composition and flavor precursor components that were significantly different between the different altitudes. The results of the VIP analysis are shown in the Appendix A, and the correlation analysis with the significant components and cupping indicators are shown in Figure 5.

Two of the seven trace elements showed significant differences at different altitudes, namely manganese and iron. The correlation analysis showed that the presence of manganese had a positive effect on coffee cupping, whereas the presence of iron had a negative effect. Although trace elements directly affected the coffee body and metal chelates were formed during the brewing process that affect coffee acidity [21,43], the differences caused by altitude mainly had a very significant impact on the aroma and flavor indicators, and also had a significant impact on balance.

Linoleic and palmitic acids are fatty acids that cause significant differences due to altitude. They had a positive impact on coffee cupping, mainly reflected in the aroma, flavor, and body, which also made the balance better. This result was also reflected in other research results [4]. This effect might stem from the natural aroma of these fatty acids and their ability to create bubbles and a creamy texture in the mouth, enhancing the aroma by capturing and holding volatile compounds [32]. Although a positive impact on body was noted, no clear trend emerged in cupping, potentially due to the high scores associated with the Pu’er coffee body itself.

Alkaloids (caffeine and trigonelline) were significantly different due to altitude, while only 3-CQA among the six different chlorogenic acids was significantly different due to altitude. Their presence negatively affected the cupping testing, affecting the aroma, flavor, aftertaste, balance, and overall quality [25,27]. Alkaloids and CGAs generally decreased with rising elevation, while their concentrations decreased with rising elevation. The coffee cupping score also supports this view.

Monosaccharides and organic acids could directly influence acidity and sweetness during cupping. According to the VIP analysis, there were three kinds of monosaccharides significantly affected by altitude, namely D-(+)-talose, D-(+)-mannose, and DL-arabinose, while there were two kinds of organic acids, namely malic acid and quinic acid. The correlation analysis showed that monosaccharides had a positive effects on coffee cupping, and only D-(+)-talose had significant effects on aroma and balance. Organic acids had a negative effect on coffee cupping, especially malic acid, with a significant effect on aroma, flavor, and balance. While quinic acid had a major negative impact, it also had a positive effect in aftertaste. It might be that the contents of quinic acid and malic acid were reduced at high altitudes, reducing the bitterness of the coffee and improving the quality of the high-altitude coffee [29,30].

The relationship between aroma and increasing altitudes was inconsistent with the levels of volatile components. The aroma index encompasses not only the fragrance of the coffee powder (volatile components) but also the aroma released after brewing [19]. The components that significantly changed with altitude resulted in a decrease in nutty and roasted aromas while enhancing sweet and caramel notes. Although the concentration of the coffee aroma decreases, the overall flavor perception improves, leading to higher scores.

## 4. Conclusions

The growing altitude was an important factor affecting the quality of coffee, which has an important impact on the cupping quality. The concentration of trace elements, alkaloids, and CGAs in coffee beans decreased with the increase in growing altitude, while the content of fatty acids increased, and the content of organic acids showed no obvious change trends. The volatile flavor components in the roasted coffee also changed with the growing altitude. This also allows the whole flavor system to reduce the nutty and roasted flavor with increasing growing altitudes, while adding the sweet and caramel aroma. High-altitude coffee also had high scores due to the increase in flavor and aroma score during coffee cupping. This also provides a scientific basis for the claim that high-altitude coffee has good quality.

Although this study has achieved the expected effects, there were still some limitations. First of all, the number of samples collected was relatively small, which may cause certain errors in the research results. Secondly, coffee quality was affected by many factors, and the change in growing environment and other factors may have a certain impact on the results. However, the results of this study were still of great significance to the evaluation, sales, and promotion of quality coffee in the Pu’er region. Of course, the next step will be to conduct relevant research on other coffee producing areas in China, in order to increase the scientific and technological connotation of coffee in Chinese producing areas.

## Figures and Tables

**Figure 1 foods-13-03842-f001:**
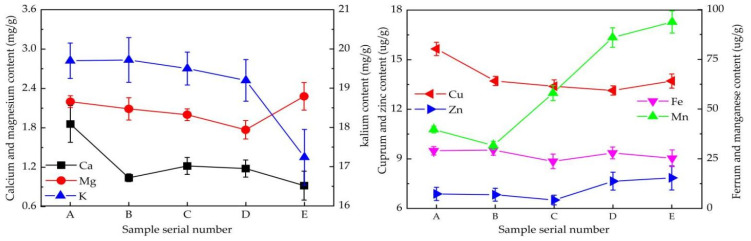
The compositions and contents of trace elements at different altitudes.

**Figure 2 foods-13-03842-f002:**
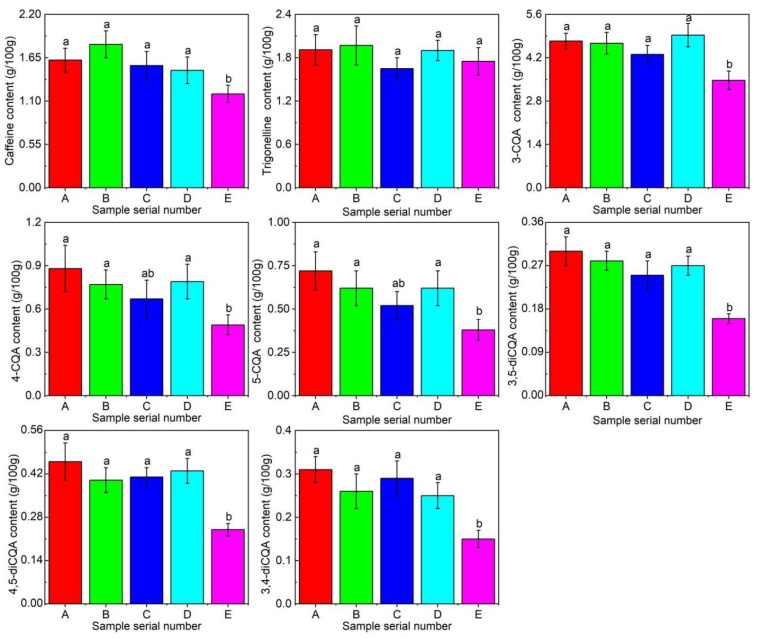
The composition and content of alkaloids and chlorogenic acids at different altitudes. (Different letters in the same line indicate significantly differences (*p* < 0.05) from each other using Duncan’s multiple comparison test).

**Figure 3 foods-13-03842-f003:**
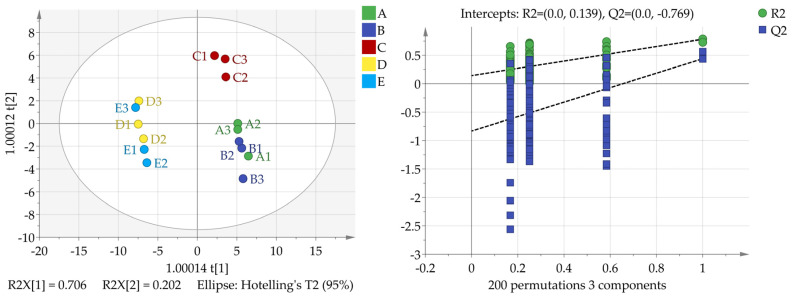
The evaluation results of the electronic senses.

**Figure 4 foods-13-03842-f004:**
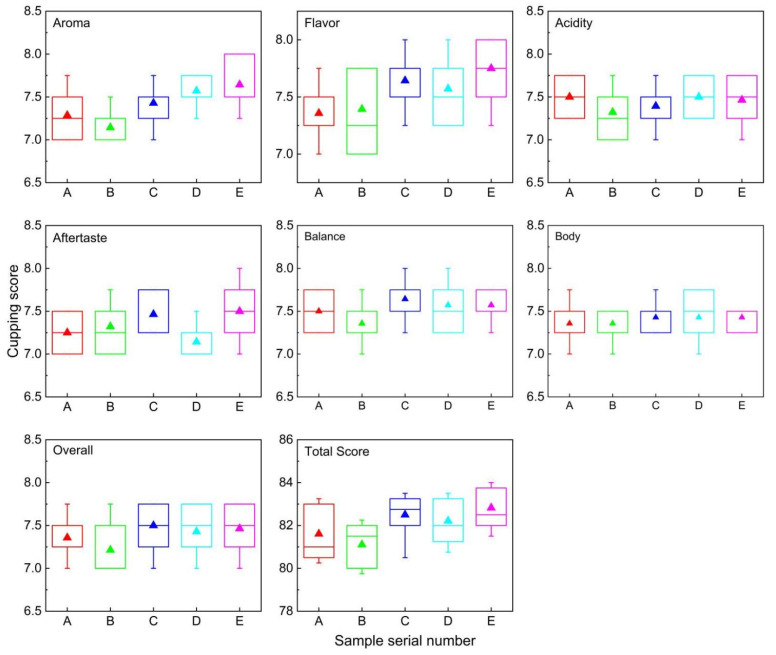
The evaluation results of coffee cupping.

**Figure 5 foods-13-03842-f005:**
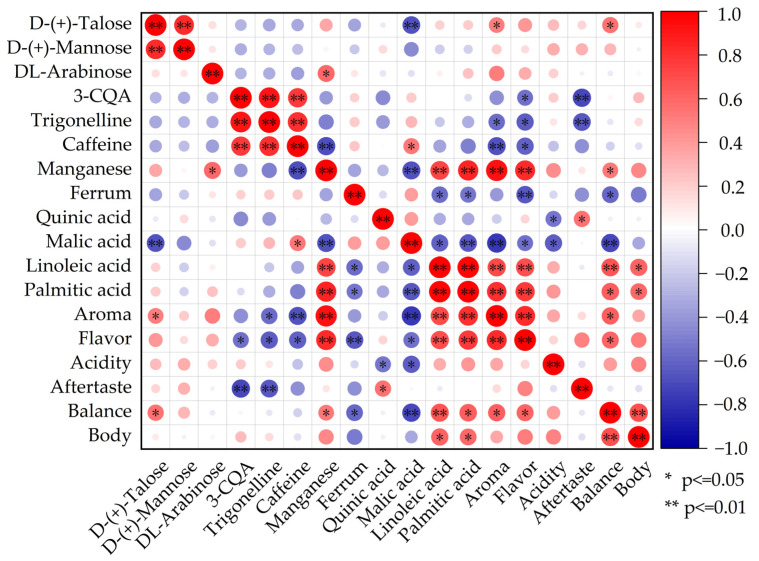
Correlation analysis of effects of flavor precursors on cupping quality.

**Table 1 foods-13-03842-t001:** Flavor precursor compositions and contents of coffee samples at different altitudes.

Type	Composition	Content (mg/g)				
		A	B	C	D	E
Organic acid	Oxalic acid	0.63 ± 0.10 ^a^	0.36 ± 0.06 ^c^	0.55 ± 0.19 ^b^	0.23 ± 0.13 ^d^	0.66 ± 0.09 ^a^
	Quinic acid	13.26 ± 0.93 ^c^	15.28 ± 0.23 ^a^	15.06 ± 0.18 ^ab^	11.31 ± 0.53 ^d^	14.84 ± 0.57 ^b^
	Acetic acid	1.23 ± 0.31 ^a^	0.97 ± 0.20 ^b^	0.13 ± 0.03 ^d^	0.20 ± 0.06 ^cd^	0.24 ± 0.09 ^c^
	Citric acid	5.76 ± 1.12 ^a^	6.28 ± 0.95 ^a^	0.10 ± 0.02 ^c^	0.95 ± 1.23 ^b^	0.08 ± 0.02 ^c^
	Malic acid	7.71 ± 0.52 ^b^	29.09 ± 4.09 ^a^	6.91 ± 1.19 ^bc^	4.67 ± 1.15 ^cd^	5.67 ± 0.07 ^c^
Fatty acid	Linoleic acid	6.55 ± 1.20 ^c^	7.54 ± 0.93 ^c^	16.41 ± 0.54 ^a^	17.94 ± 1.16 ^a^	13.76 ± 1.08 ^b^
	Palmitic acid	3.78 ± 0.56 ^c^	4.12 ± 0.47 ^c^	7.55 ± 0.19 ^b^	8.99 ± 0.53 ^a^	7.68 ± 0.90 ^b^
	Stearic acid	1.18 ± 0.19 ^c^	1.46 ± 0.35 ^c^	3.06 ± 0.26 ^ab^	3.56 ± 0.62 ^a^	2.75 ± 0.48 ^b^
	Oleic acid	1.02 ± 0.22 ^c^	1.28 ± 0.33 ^c^	2.91 ± 0.32 ^a^	2.88 ± 0.80 ^ab^	2.72 ± 0.73 ^b^
	Arachidic acid	0.56 ± 0.08 ^c^	0.80 ± 0.24 ^bc^	1.72 ± 0.27 ^a^	1.59 ± 0.65 ^a^	1.38 ± 0.42 ^ab^
	Linolenic acid	0.16 ± 0.06 ^b^	0.18 ± 0.02 ^b^	0.52 ± 0.05 ^a^	0.48 ± 0.18 ^a^	0.37 ± 0.13 ^a^
Monosaccharide	L-(+)-Threose	28.40 ± 3.28 ^a^	12.86 ± 1.40 ^d^	14.42 ± 0.97 ^c^	27.57 ± 5.26 ^ab^	25.45 ± 4.14 ^b^
	DL-Arabinose	39.10 ± 3.24 ^c^	43.29 ± 4.27 ^c^	21.34 ± 4.34 ^d^	51.80 ± 4.74 ^b^	68.60 ± 4.63 ^a^
	D-(+)-Talose	144.84 ± 21.52 ^ab^	68.99 ± 6.11 ^d^	134.20 ± 8.74 ^b^	99.74 ± 16.20 ^c^	155.43 ± 15.90 ^a^
	D-(+)-Galactose	24.23 ± 1.94 ^a^	8.09 ± 1.19 ^d^	14.90 ± 1.87 ^b^	11.95 ± 2.42 ^c^	24.56 ± 2.93 ^a^
	D-(+)-Mannose	80.47 ± 4.10 ^a^	25.66 ± 2.76 ^d^	51.47 ± 6.08 ^c^	24.16 ± 1.58 ^d^	73.02 ± 1.71 ^b^

Note: Different letters in the same line indicate significantly differences (*p* < 0.05) from each other using Duncan’s multiple comparison test.

**Table 2 foods-13-03842-t002:** Components that significantly change with increasing growing altitudes.

Chemical Compounds	VIP	Tendency	Flavor Characteristic
Pyrazine, methyl-	6.19	decrease	Nutty, roasted, sweet, chocolate
2-furanmethanol	5.08	decrease	Mild, slightly caramellic
Furfural	3.13	increase	Sweet, bread-like, caramellic
Pyridine	2.77	decrease	Pungent, penetrating, diffusive
Pyrazine, 2,6-dimethyl-	2.24	decrease	Sweet, fried, nutty, roasted, chocolate
Pyrazine, 2,5-dimethyl-	1.78	decrease	Roasted, nutty, grassy, chocolate
Pyrazine, 2-ethyl-6-methyl-	1.76	decrease	Roasted hazelnut-like
2-furancarboxaldehyde, 5-methyl-	1.70	increase	Sweet-spicy, warm, caramellic
Pyrazine, 2-ethyl-3-methyl-	1.41	decrease	Nutty, roasted
Pyrazine, 3-ethyl-2,5-dimethyl-	1.36	decrease	Hazelnut, roasty, earthy
Pyrazine, ethyl-	1.25	decrease	Nutty, roasted, buttery, rum

## Data Availability

The original contributions presented in the study are included in the article/Appendix A, further inquiries can be directed to the corresponding author.

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
