# Peer review of "The Growing Altitude Influences the Flavor Precursors, Sensory Characteristics and Cupping Quality of the Pu’er Coffee Bean"

_foods, 2024, doi:10.3390/foods13233842_

Round 1
Reviewer 1 Report
Comments and Suggestions for Authors
The article may be a contribution on the effect of altitude on the quality of coffee beans, the methodology used is correct, however the conclusions must agree with the scope of the study
A further review of the bibliographical background is required
Flavor precursors in coffee are not identified in the introduction. Are all the groups of compounds studied flavor precursors? Or, are some compounds actually responsible for flavor and others are flavor precursors? This question should be made clear in the introduction.
Reference N°[4 ], line 38. The reference does not correspond, it is about the market in Brazil and not about the quality of coffee in China.
Line 43, must include reference
Line 45, could expand on the results obtained in reference No. 5
Lines 45 and 46 indicate the altitude in feet, it is suggested to include the altitude in meters, to compare the heights because the text includes meters and feet
Line 53-54 should indicate the precursors of the flavor to be studied and their effect on the final quality
The determination of trace elements, what happens to the territory? If the regions are different, the composition of trace elements is also different
Materials and methods
The origin of the samples is not clearly identified, it would be a contribution to include a geo-map locating the place of origin of the samples, it indicates different regions, which generates confusion
Table 1 indicates whether the significant differences correspond to the row or the column
Because the temperature decreases with altitude, it is convenient to indicate the temperature record to establish the differences between these in each of the sampling places
In the discussion, the results should be integrated, for example, the concentration of monosaccharides and the presence of volatile compounds and sweet perception
In addition, the difference or similarities of your results with results reported in the bibliography should be indicated.
Author Response
Dear Reviewer ,
Thank you for your useful comments and suggestions on the structure of our manuscript. We have modified the manuscript accordingly, and detailed corrections are listed below point by point:
1) The article may be a contribution on the effect of altitude on the quality of coffee beans, the methodology used is correct, however the conclusions must agree with the scope of the study
Thanks for your reminding. I have rewritten the conclusion to make it more consistent with the research scope.
2) Flavor precursors in coffee are not identified in the introduction. Are all the groups of compounds studied flavor precursors? Or, are some compounds actually responsible for flavor and others are flavor precursors? This question should be made clear in the introduction.
Thanks for your reminder. I have added an introduction to the flavor precursor components to the introduction.
3) Reference N [4], line 38. The reference does not correspond, it is about the market in Brazil and not about the quality of coffee in China.
Although the reference is about the coffee market in Brazil, it also mentions Chinese coffee, so I use it as a citation. I removed this reference from the manuscript to avoid unnecessary misunderstandings. The introduction of coffee market in China is reported in China daily, and the specific connection is as follows: https://www.chinadaily.com.cn/a/202204/27/WS6268a03da310fd2b29e59924.html
4) Line 43, must include reference
References have been added. The reference is a review with a lot of discussion about altitude and coffee quality.
5) Line 45, could expand on the results obtained in reference No. 5
Thank you for your valuable advice, and I have thought about this proposal for a long time. Reference No.5 is that the effect of altitude on the biochemical composition and quality of coffee beans can be affected by the shade and postharvest processing methods. The literature mainly elucidates the main and two-way interactive effects of altitude, shade and post-harvest processing on the biochemical composition and quality attributes (physical quality and acidity) of coffee beans. It is fundamentally different from the present study. In the introduction section, what I mean is not quite consistent with the results of this reference, so I think it is inappropriate to extend the results of the reference in the reference. If you still feel the need to extend the study findings, I can add the results to the manuscript.
6) Lines 45 and 46 indicate the altitude in feet, it is suggested to include the altitude in meters, to compare the heights because the text includes meters and feet
Thanks for your advice, I have changed the feet to meters
7) Line 53-54 should indicate the precursors of the flavor to be studied and their effect on the final quality
Sorry, this could be a writing mistake. Corrected.
8) The determination of trace elements, what happens to the territory? If the regions are different, the composition of trace elements is also different
Trace elements are one of the important components of coffee beans, and their content and composition are affected by the planting soil. Therefore, trace elements also become an important evidence for the traceability of coffee sources. Trace elements are also one of the factors that affect the quality of coffee cupping. Therefore, the manuscript takes trace elements as the important indicators when measuring the quality of coffee cupping.
9) The origin of the samples is not clearly identified, it would be a contribution to include a geo-map locating the place of origin of the samples, it indicates different regions, which generates confusion
I think you are right. Although the geographical map of the location indicates the location where the sample was collected, it is too abstract, so I changed it to a specific location. Village is the smallest administrative area unit in China, its area is often small. Therefore, the sample collection location can be accurately determined through this location.
10) Table 1 indicates whether the significant differences correspond to the row or the column
The significant differences in Table 1 correspond to the rows.
11) Because the temperature decreases with altitude, it is convenient to indicate the temperature record to establish the differences between these in each of the sampling places
Coffee samples were collected in the same village, and the impact of factors such as soil and climate on coffee samples would be relatively low. Coffee cherries are usually picked during the day and processed in the afternoon or night, which can keep the same quality of the coffee as much as possible. Although the temperature is affected by the altitude, it is also quite different at different times of the day, and it seems less scientific to use the temperature as a measure.
12) In the discussion, the results should be integrated, for example, the concentration of monosaccharides and the presence of volatile compounds and sweet perception
Thank you for your valuable comments. I thought deeply about your suggestions. Volatile components are perceived by the sense of smell, while components such as monosaccharides are perceived by taste. The combination of the two is not very easy to achieve. Of course, the flavor description is also carried out in the process of coffee cupping, which is subjective to a certain extent, so I did not add this part of data to the manuscript. This part of the data can be verified with volatile components and complement each other.
13) In addition, the difference or similarities of your results with results reported in the bibliography should be indicated.
Thanks for your valuable comments. I have made appropriate corrections in the manuscript to accurately describe the differences or similarities between the results and the references.
The manuscript has been resubmitted to journal. We look forward to your positive response. Let us know if you have better suggestions.
Sincerely,
Rongsuo Hu

Reviewer 2 Report
Comments and Suggestions for Authors
The authors provide interesting research into coffee from various altitudes. My major concern would be the control of all factors besides altitude, which is not made quite clear in the methods section. For example, it is not even stated that the same coffee variety was used in all experiments. It is also not clear if the roasting curve was the same for all roastings. It is also questionable when the time following roasting was not standardized, e.g. a coffee after 48 h vs. 1 week following roasting probably has higher taste differences from the deviations in maturation. All these steps must be clarified in the methods section.
Finally, the discussion appears to lack a limitations section, e.g. low number of samples, questionable transferrability, comparisons barely significant (e.g. the standard deviations are quite high). The statistical treatment might be improved, e.g. by stating p-valued for all claims of significance.
Some further corrections:
- affiliations are missing on page 1
- section 2.1.2: roasting curve could be provided (why is there a difference between 618 and 628 s of roasting, was the process somehow standardized?)
- Line 71: provide coffee species and variety
- Line 95: were roasted samples analyzed?
- Line 165: is there a µm missing for PDMS?
- Line 197: why was not a stabile maturation degree investigated after roasting, e.g. minimum of 2 weeks?
- Line 219: delete irrelevant health claims
Author Response
Dear Reviewer ,
Thank you for your useful comments and suggestions on the structure of our manuscript. We have modified the manuscript accordingly, and detailed corrections are listed below point by point:
1) The authors provide interesting research into coffee from various altitudes. My major concern would be the control of all factors besides altitude, which is not made quite clear in the methods section. For example, it is not even stated that the same coffee variety was used in all experiments. It is also not clear if the roasting curve was the same for all roastings. It is also questionable when the time following roasting was not standardized, e.g. a coffee after 48 h vs. 1 week following roasting probably has higher taste differences from the deviations in maturation. All these steps must be clarified in the methods section.
Coffee collections are made in the same village where they have the same growth environment and climatic conditions. In this way, the influence of all factors except altitude on coffee can be reduced as much as possible to ensure the accuracy of the experiment.
The planting time and variety of coffee are the same. The coffee is Coffea arabica L, the planting time is 4 years, and the variety is catimor CIFC7963 (F6).
The roasting curves are the same for all of the samples. The roasting process has a random error, which causes a slight deviation in the roasting time.
The sample flavor detection and sensory detection times are consistent, both on the sixth day after the sample is baked, which is reflected in the method.
The times of flavor detection and sensory evaluation were consistent, both on the sixth day after the samples was roasted, which is reflected in 2.4.2.
2) Finally, the discussion appears to lack a limitations section, e.g. low number of samples, questionable transferrability, comparisons barely significant (e.g. the standard deviations are quite high). The statistical treatment might be improved, e.g. by stating p-valued for all claims of significance.
Thanks for your valuable comments. I have added restrictive descriptions to the conclusion section of the manuscript.
Statistical processing has carried out significance analysis on the data. Through VIP processing, we can screen out the indicators of the significant influence on altitude, and then carry out correlation analysis. This can better highlight the influence of detection indicators on altitude. Relevant processing results are in the supplementary documents, please review.
3) affiliations are missing on page 1
Foods remove affiliation during peer review, which is relatively fair. It will be added when official publication.
4) section 2.1.2: roasting curve could be provided (why is there a difference between 618 and 628 s of roasting, was the process somehow standardized?)
The roasting curve is the curve of temperature and time, which can provide a reference for coffee roasting and ensure the stability and consistency of roasting samples. The roasting curves have been provided, and are placed in the supplementary file.
Due to random errors in coffee roasting, the time can deviate by several seconds at the desired temperature.
5) Line 71: provide coffee species and variety
The experimental sample is Coffea arabica L, the variety is catimor CIFC7963 (F6). This has been supplementary illustrated in the manuscript.
6) Line 95: were roasted samples analyzed?
Yes, the volatile component analysis was made from roasted samples.
7) Line 165: is there a µm missing for PDMS?
Thanks for the reminder. The μ m is already present in line 165.
8) Line 197: why was not a stabile maturation degree investigated after roasting, e.g. minimum of 2 weeks?
The sensory evaluation of roasted coffee was performed on the sixth day of the roasted sample. This is described in 2.4.2. We did a cupping analysis before the sensory evaluation and had the best sensory score on sixth day.
9) Line 219: delete irrelevant health claims
Thanks for your reminder, I have removed the health statement.
The manuscript has been resubmitted to journal. We look forward to your positive response. Let us know if you have better suggestions.
Sincerely,
Rongsuo Hu

Round 2
Reviewer 1 Report
Comments and Suggestions for Authors
The inclusion of the suggested corrections is appreciated, which are correct, however the conclusion corresponds to a discussion/conclusion
Author Response
1) The inclusion of the suggested corrections is appreciated, which are correct, however the conclusion corresponds to a discussion/conclusion
✓ Thank you for your valuable advice. I rewrote the conclusion of the manuscript. Please review
